# Submodular meets Structured: Finding Diverse Subsets in Exponentially-Large Structured Item Sets

**Adarsh Prasad**
UT Austin
adarsh@cs.utexas.edu

**Stefanie Jegelka**
UC Berkeley
stefje@eecs.berkeley.edu

**Dhruv Batra**
Virginia Tech
dbatra@vt.edu

## Abstract

To cope with the high level of ambiguity faced in domains such as Computer Vision or Natural Language processing, robust prediction methods often search for a *diverse set* of high-quality candidate solutions or *proposals*. In structured prediction problems, this becomes a daunting task, as the solution space (image labelings, sentence parses, *etc.*) is exponentially large. We study greedy algorithms for finding a diverse subset of solutions in structured-output spaces by drawing new connections between submodular functions *over combinatorial item sets* and High-Order Potentials (HOPs) studied for graphical models. Specifically, we show via examples that when marginal gains of submodular diversity functions allow structured representations, this enables efficient (sub-linear time) approximate maximization by reducing the greedy augmentation step to inference in a factor graph with appropriately constructed HOPs. We discuss benefits, trade-offs, and show that our constructions lead to significantly better proposals.

## 1 Introduction

Many problems in Computer Vision, Natural Language Processing and Computational Biology involve mappings from an input space $\mathcal{X}$ to an exponentially large space $\mathcal{Y}$ of *structured outputs*. For instance, $\mathcal{Y}$ may be the space of all segmentations of an image with $n$ pixels, each of which may take $L$ labels, so $|\mathcal{Y}| = L^n$. Formulations such as Conditional Random Fields (CRFs) [24], Max-Margin Markov Networks (M³N) [31], and Structured Support Vector Machines (SSVMs) [32] have successfully provided principled ways of scoring all solutions $\mathbf{y} \in \mathcal{Y}$ and predicting the *single* highest scoring or maximum *a posteriori* (MAP) configuration, by exploiting the factorization of a structured output into its constituent "parts".

In a number of scenarios, the posterior $\mathbf{P}(\mathbf{y}|\mathbf{x})$ has several modes due to ambiguities, and we seek not only a single best prediction but a *set* of good predictions:
(1) **Interactive Machine Learning.** Systems like Google Translate (for machine translation) or Photoshop (for interactive image segmentation) solve structured prediction problems that are often ambiguous ("what did the user really mean?"). Generating a small set of relevant candidate solutions for the user to select from can greatly improve the results.
(2) **M-Best hypotheses in cascades.** Machine learning algorithms are often cascaded, with the output of one model being fed into another [33]. Hence, at the initial stages it is not necessary to make a single perfect prediction. We rather seek a set of *plausible* predictions that are subsequently re-ranked, combined or processed by a more sophisticated mechanism.
In both scenarios, we ideally want a small set of $M$ *plausible* (*i.e.*, high scoring) but *non-redundant* (*i.e.*, diverse) structured-outputs to hedge our bets.

**Submodular Maximization and Diversity.** The task of searching for a diverse high-quality subset of items from a ground set $V$ has been well-studied in information retrieval [5], sensor placement [22], document summarization [26], viral marketing [17], and robotics [10]. Across these domains, *submodularity* has emerged as an a fundamental and practical concept – a property of functions for measuring diversity of a subset of items. Specifically, a set function $F : 2^V \to \mathbb{R}$ is submodular if its *marginal gains*, $F(a|S) \equiv F(S \cup a) - F(S)$ are decreasing, *i.e.* $F(a|S) \geq F(a|T)$

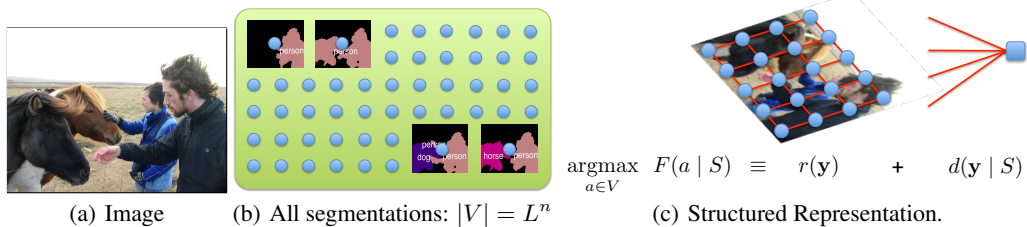

| (a) Image | (b) All segmentations: $|V| = L^n$ | (c) Structured Representation. |

$$\operatorname*{argmax}_{a \in V} \ F(a \mid S) \ \equiv \ r(\mathbf{y}) \ + \ d(\mathbf{y} \mid S)$$

Figure 1: (a) input image; (b) space of all possible object segmentations / labelings (each item is a segmentation); (c) we convert the problem of finding the item with the highest marginal gain $F(a|S)$ to a MAP inference problem in a factor graph over base variables $\mathbf{y}$ with an appropriately defined HOP.

for all $S \subseteq T$ and $a \notin T$. In addition, if $F$ is *monotone*, *i.e.*, $F(S) \leq F(T), \ \forall S \subseteq T$, then a simple greedy algorithm (that in each iteration $t$ adds to the current set $S^t$ the item with the largest marginal gain $F(a|S^t)$) achieves an approximation factor of $(1 - \frac{1}{e})$ [27]. This result has had significant practical impact [21]. Unfortunately, if the number of items $|V|$ is exponentially large, then *even a single linear scan for greedy augmentation is infeasible*.

In this work, we study conditions under which it is feasible to greedily maximize a submodular function over an exponentially large ground set $V = \{v_1, \ldots, v_N\}$ whose elements are *combinatorial objects*, i.e., labelings of a *base set* of $n$ variables $\mathbf{y} = \{y_1, y_2, \ldots, y_n\}$. For instance, in image segmentation, the base variables $y_i$ are pixel labels, and each item $a \in V$ is a particular labeling of the pixels. Or, if each base variable $y_e$ indicates the presence or absence of an edge $e$ in a graph, then each item may represent a spanning tree or a maximal matching. Our goal is to find a set of $M$ plausible and diverse configurations *efficiently*, *i.e.* in time sub-linear in $|V|$ (ideally scaling as a low-order polynomial in $\log |V|$). We will assume $F(\cdot)$ to be monotone submodular, nonnegative and normalized ($F(\emptyset) = 0$), and base our study on the greedy algorithm. As a running example, we focus on pixel labeling, where each base variable takes values in a set $[L] = \{1, \ldots, L\}$ of labels.

**Contributions.** Our principal contribution is a conceptual one. We observe that *marginal gains of a number of submodular functions allow structured representations*, and this enables efficient greedy maximization over exponentially large ground sets – by reducing the greedy augmentation step to a MAP inference query in a discrete factor graph augmented with a suitably constructed *High-Order Potential* (HOP). Thus, our work draws new connections between two seemingly disparate but highly related areas in machine learning – submodular maximization and inference in graphical models with structured HOPs. As specific examples, we construct submodular functions for three different, task-dependent definitions of diversity, and provide reductions to three different HOPs for which efficient inference techniques have already been developed. Moreover, we present a generic recipe for constructing such submodular functions, which may be "plugged" with efficient HOPs discovered in future work. Our empirical contribution is an efficient algorithm for producing a set of image segmentations with significantly higher *oracle accuracy*[1] than previous works. The algorithm is general enough to transfer to other applications. Fig. 1 shows an overview of our approach.

**Related work: generating multiple solutions.** Determinental Point Processesare an elegant probabilistic model over sets of items with a preference for diversity. Its generalization to a structured setting [23] assumes a tree-structured model, an assumption that we do not make. Guzman-Rivera *et al*. [14, 15] learn a set of $M$ models, each producing one solution, to form the set of solutions. Their approach requires access to the learning sub-routine and repeated re-training of the models, which is not always possible, as it may be expensive or proprietary. We assume to be given a single (pre-trained) model from which we must generate multiple diverse, good solutions. Perhaps the closest to our setting are recent techniques for finding diverse $M$-best solutions [2, 28] or modes [7, 8] in graphical models. While [7] and [8] are inapplicable since they are restricted to chain and tree graphs, we compare to other baselines in Section 3.2 and 4.

## 1.1 Preliminaries and Notation

We select from a ground set $V$ of $N$ items. Each item is a labeling $\mathbf{y} = \{y_1, y_2, \ldots, y_n\}$ of $n$ base variables. For clarity, we use non-bold letters $a \in V$ for items, and boldface letters $\mathbf{y}$ for base set configurations. Uppercase letters refer to functions over the ground set items $F(a|A), R(a|A), D(a|A)$, and lowercase letters to functions over base variables $f(\mathbf{y}), r(\mathbf{y}), d(\mathbf{y})$.

Formally, there is a bijection $\phi : V \mapsto [L]^m$ that maps items $a \in V$ to their representation as base variable labelings $\mathbf{y} = \phi(a)$. For notational simplicity, we often use $\mathbf{y} \in S$ to mean $\phi^{-1}(\mathbf{y}) \in S$, *i.e.* the item corresponding to the labeling $\mathbf{y}$ is present in the set $S \subseteq V$. We write $\ell \in \mathbf{y}$ if the label $\ell$ is used in $\mathbf{y}$, *i.e.* $\exists j$ s.t. $y_j = \ell$. For a set $c \subseteq [n]$, we use $y_c$ to denote the tuple $\{y_i \mid i \in c\}$. Our goal to find an ordered set or *list* of items $S \subseteq V$ that maximizes a scoring function $F$. Lists generalize the notation of sets, and allow for reasoning of item order and repetitions. More details about list vs set prediction can be found in [29, 10].

**Scoring Function.** We trade off the relevance and diversity of list $S \subseteq V$ via a scoring function $F : 2^V \to \mathbb{R}$ of the form

$$F(S) = R(S) + \lambda D(S), \qquad (1)$$

where $R(S) = \sum_{a \in S} R(a)$ is a modular nonnegative relevance function that aggregates the quality of all items in the list; $D(S)$ is a monotone normalized submodular function that measure the diversity of items in $S$; and $\lambda \geq 0$ is a trade-off parameter. Similar objective functions were used e.g. in [26]. They are reminiscent of the general paradigm in machine learning of combining a loss function that measures quality (*e.g.* training error) and a regularization term that encourages desirable properties (*e.g.* smoothness, sparsity, or "diversity").

**Submodular Maximization.** We aim to find a list $S$ that maximizes $F(S)$ subject to a cardinality constraint $|S| \leq M$. For monotone submodular $F$, this may be done via a greedy algorithm that starts out with $S^0 = \emptyset$, and iteratively adds the next best item:

$$S^t = S^{t-1} \cup a^t, \qquad a^t \in \operatorname{argmax}_{a \in V} F(a \mid S^{t-1}). \qquad (2)$$

The final solution $S^M$ is within a factor of $(1 - \frac{1}{e})$ of the optimal solution $S^*$: $F(S^M) \geq (1 - \frac{1}{e})F(S^*)$ [27]. The computational bottleneck is that in each iteration, we must find the item with the largest marginal gain. Clearly, if $|V|$ has exponential size, we cannot touch each item even once. Instead, we propose "augmentation sub-routines" that exploit the structure of $V$ and maximize the marginal gain by solving an optimization problem over the base variables.

## 2 Marginal Gains in Configuration Space

To solve the greedy augmentation step via optimization over $\mathbf{y}$, we transfer the marginal gain from the world of items to the world of base variables and derive functions on $\mathbf{y}$ from $F$:

$$\underbrace{F(\phi^{-1}(\mathbf{y}) \mid S)}_{f(\mathbf{y}|S)} = \underbrace{R(\phi^{-1}(\mathbf{y}))}_{r(\mathbf{y})} + \lambda \underbrace{D(\phi^{-1}(\mathbf{y}) \mid S)}_{d(\mathbf{y}|S)}. \qquad (3)$$

Maximizing $F(a|S)$ now means maximizing $f(\mathbf{y}|S)$ for $\mathbf{y} = \phi(a)$. This can be a hard combinatorial optimization problem in general. However, as we will see, there is a broad class of useful functions $F$ for which $f$ inherits exploitable structure, and $\operatorname{argmax}_{\mathbf{y}} f(\mathbf{y}|S)$ can be solved efficiently, exactly or at least approximately.

**Relevance Function.** We use a structured relevance function $R(a)$ that is the score of a factor graph defined over the base variables $\mathbf{y}$. Let $G = (\mathcal{V}, \mathcal{E})$ be a graph defined over $\{y_1, y_2, \ldots, y_n\}$, *i.e.* $\mathcal{V} = [n]$, $\mathcal{E} \subseteq \binom{\mathcal{V}}{2}$. Let $\mathcal{C} = \{C \mid C \subseteq \mathcal{V}\}$ be a set of cliques in the graph, and let $\theta_C : [L]^{|C|} \mapsto \mathbb{R}$ be the log-potential functions (or factors) for these cliques. The quality of an item $a = \phi^{-1}(\mathbf{y})$ is then given by $R(a) = r(\mathbf{y}) = \sum_{C \in \mathcal{C}} \theta_C(y_C)$. For instance, with only node and edge factors, this quality becomes $r(\mathbf{y}) = \sum_{p \in \mathcal{V}} \theta_p(y_p) + \sum_{(p,q) \in \mathcal{E}} \theta_{pq}(y_p, y_q)$. In this model, finding the *single* highest quality item corresponds to maximum a posteriori (MAP) inference in the factor graph.

Although we refer to terms with probabilistic interpretations such as "MAP", we treat our relevance function as output of an *energy-based model* [25] such as a Structured SVM [32]. For instance, $r(\mathbf{y}) = \sum_{C \in \mathcal{C}} \theta_C(y_C) = \mathbf{w}^\intercal \psi(\mathbf{y})$ for parameters $\mathbf{w}$ and feature vector $\psi(\mathbf{y})$. Moreover, we assume that the relevance function $r(\mathbf{y})$ is nonnegative[2]. This assumption ensures that $F(\cdot)$ is monotone. If $F$ is non-monotone, algorithms other than the greedy are needed [4, 12]. We leave this generalization for future work. In most application domains the relevance function is learned from data and thus our positivity assumption is not restrictive – *one can simply learn a positive relevance function*. For instance, in SSVMs, the relevance weights are learnt to maximize the margin between the correct labeling and all incorrect ones. We show in the supplement that SSVM parameters that assign nonnegative scores to all labelings achieve exactly the same hinge loss (and thus the same generalization error) as without the nonnegativity constraint.

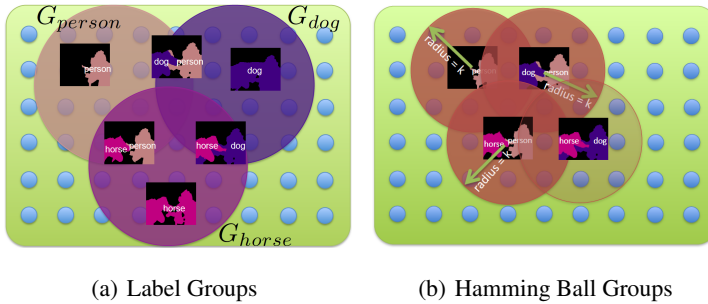

| (a) Label Groups | (b) Hamming Ball Groups |

Figure 2: Diversity via groups: (a) groups defined by the presence of labels (*i.e.* #groups = $L$); (b) groups defined by Hamming balls around each item/labeling (*i.e.* #groups = $L^n$). In each case, diversity is measured by how many groups are covered by a new item. See text for details.

## 3 Structured Diversity Functions

We next discuss a general recipe for constructing monotone submodular diversity functions $D(S)$, and for reducing their marginal gains to structured representations over the base variables $d(\mathbf{y}|S)$. Our scheme relies on constructing *groups* $G_i$ that cover the ground set, *i.e.* $V = \bigcup_i G_i$. These groups will be defined by task-dependent characteristics – for instance, in image segmentation, $G_\ell$ can be the set of all segmentations that contain label $\ell$. The groups can be overlapping. For instance, if a segmentation $\mathbf{y}$ contains pixels labeled "grass" and "cow", then $\mathbf{y} \in G_{\text{grass}}$ and $\mathbf{y} \in G_{\text{cow}}$.

**Group Coverage: Count Diversity.** Given $V$ and a set of groups $\{G_i\}$, we measure the diversity of a list $S$ in terms of its *group coverage*, *i.e.*, the number of groups covered jointly by items in $S$:

$$D(S) = \Big| \big\{ i \mid G_i \cap S \neq \emptyset \big\} \Big|, \tag{4}$$

where we define $G_i \cap S$ as the intersection of $G_i$ with the set of unique items in $S$. It is easy to show that this function is monotone submodular. If $G_\ell$ is the group of all segmentations that contain label $\ell$, then the diversity measure of a list of segmentations $S$ is the number of object labels that appear in any $a \in S$. The marginal gain is the number of new groups covered by $a$:

$$D(a \mid S) = \Big| \big\{ i \mid a \in G_i \text{ and } S \cap G_i = \emptyset \big\} \Big|. \tag{5}$$

Thus, the greedy algorithm will try to find an item/segmentation that belongs to as many as yet unused groups as possible.

**Group Coverage: General Diversity.** More generally, instead of simply counting the number of groups covered by $S$, we can use a more refined decay

$$D(S) = \sum_i h\big( \big| G_i \cap S \big| \big). \tag{6}$$

where $h$ is any nonnegative nondecreasing concave scalar function. This is a sum of submodular functions and hence submodular. Eqn. (4) is a special case of Eqn. (6) with $h(y) = \min\{1, y\}$. Other possibilities are $\sqrt{\cdot}$, or $\log(1 + \cdot)$. For this general definition of diversity, the marginal gain is

$$D(a \mid S) = \sum_{i : G_i \ni a} \left[ h\big( 1 + \big| G_i \cap S \big| \big) - h\big( \big| G_i \cap S \big| \big) \right]. \tag{7}$$

Since $h$ is concave, the gain $h\big( 1 + \big| G_i \cap S \big| \big) - h\big( \big| G_i \cap S \big| \big)$ decreases as $S$ becomes larger. Thus, the marginal gain of an item $a$ is proportional to how rare each group $G_i \ni a$ is in the list $S$.

In each step of the greedy algorithm, we maximize $r(\mathbf{y}) + \lambda d(\mathbf{y}|S)$. We already established a structured representation of $r(\mathbf{y})$ via a factor graph on $\mathbf{y}$. In the next few subsections, we specify three example definitions of groups $G_i$ that instantiate three diversity functions $D(S)$. For each $D(S)$, we show how the marginal gains $D(a|S)$ can be expressed as a specific High-Order Potential (HOP) $d(\mathbf{y}|S)$ in the factor graph over $\mathbf{y}$. These HOPs are known to be efficiently optimizable, and hence we can solve the augmentation step efficiently. Table 1 summarizes these connections.

**Diversity and Parsimony.** If the groups $G_i$ are overlapping, some $\mathbf{y}$ can belong to many groups simultaneously. While such a $\mathbf{y}$ may offer an immediate large gain in diversity, in many applications it is more natural to seek a small list of *complementary* labelings rather than having all labels occur in the same $\mathbf{y}$. For instance, in image segmentation with groups defined by label presence (Sec. 3.1), natural scenes are unlikely to contain many labels at the same time. Instead, the labels should be spread *across* the selected labelings $\mathbf{y} \in S$. Hence, we include a *parsimony factor* $p(\mathbf{y})$ that biases towards simpler labelings $\mathbf{y}$. This term is a modular function and does not affect the diversity functions directly. We next outline some example instantiations of the functions (4) and (6).

| | Groups ($G_i$) | Higher Order Potentials |
| --- | --- | --- |
| Section 3.1 | Labels | Label Cost |
| Supplement | Label Transitions | Co-operative Cuts |
| Section 3.2 | Hamming Balls | Cardinality Potentials |

Table 1: Different diversity functions and corresponding HOPs.

## 3.1 Diversity of Labels

For the first example, let $G_\ell$ be the set of all labelings $\mathbf{y}$ containing the label $\ell$, *i.e.* $\mathbf{y} \in G_\ell$ if and only if $y_j = \ell$ for some $j \in [n]$. Such a diversity function arises in multi-class image segmentation – if the highest scoring segmentation contains "sky" and "grass", then we would like to add complementary segmentations that contain an unused class label, say "sheep" or "cow".

**Structured Representation of Marginal Gains.** The marginal gain for this diversity function turns out to be a HOP called *label cost* [9]. It penalizes each label that occurs in a previous segmentation. Let $\mathtt{lcount}_S(\ell)$ be the number of segmentations in $S$ that contain label $\ell$. In the simplest case of coverage diversity (4), the marginal gain provides a constant reward for every as yet unseen label $\ell$:

$$d(\mathbf{y} \mid S) = \left| \left\{ \ell \mid \mathbf{y} \in G_\ell, S \cap G_\ell = \emptyset \right\} \right| = \sum_{\ell \in \mathbf{y}, \mathtt{lcount}_S(\ell)=0} 1. \qquad (8)$$

For the general group coverage diversity (6), the gain becomes

$$d(\mathbf{y}|S) = \sum_{\ell: G_\ell \ni \mathbf{y}} \left[ h\big(1 + |G_\ell \cap S|\big) - h\big(|G_\ell \cap S|\big) \right] = \sum_{\ell \in \mathbf{y}} \left[ h\big(1 + \mathtt{lcount}_S(\ell)\big) - h\big(\mathtt{lcount}_S(\ell)\big) \right].$$

Thus, $d(\mathbf{y}|S)$ rewards the presence of a label $\ell$ in $\mathbf{y}$ by an amount proportional to how rare $\ell$ is in the segmentations already chosen in $S$. The parsimony factor in this setting is $p(\mathbf{y}) = \sum_{\ell \in \mathbf{y}} c(\ell)$. In the simplest case, $c(\ell) = -1$, *i.e.* we are charged a constant for every label used in $\mathbf{y}$.

With this type of diversity (and parsimony terms), the greedy augmentation step is equivalent to performing MAP inference in a factor graph augmented with label reward HOPs: $\mathrm{argmax}_\mathbf{y}\, r(\mathbf{y}) + \lambda(d(\mathbf{y} \mid S) + p(\mathbf{y}))$. Delong et al. [9] show how to perform approximate MAP inference with such label costs via an extension to the standard $\alpha$-expansion [3] algorithm.

**Label Transitions.** Label Diversity can be extended to reward not just the presence of previously unseen labels, but also the presence of previously unseen *label transitions* (*e.g.*, a person in front of a car or a person in front of a house). Formally, we define one group $G_{\ell,\ell'}$ per label pair $\ell, \ell'$, and $\mathbf{y} \in G_{\ell,\ell'}$ if it contains two adjacent variables $y_i, y_j$ with labels $y_i = \ell, y_j = \ell'$. This diversity function rewards the presence of a label pair $(\ell, \ell')$ by an amount proportional to how rare this pair is in the segmentations that are part of $S$. For such functions, the marginal gain $d(\mathbf{y}|S)$ becomes a HOP called *cooperative cuts* [16]. The inference algorithm in [19] gives a fully polynomial-time approximation scheme for any nondecreasing, nonnegative $h$, and the exact gain maximizer for the count function $h(y) = \min\{1, y\}$. Further details may be found in the supplement.

## 3.2 Diversity via Hamming Balls

The label diversity function simply rewarded the presence of a label $\ell$, irrespective of which or how many variables $y_i$ were assigned that label. The next diversity function rewards a large *Hamming distance* $\mathrm{Ham}(\mathbf{y}^1, \mathbf{y}^2) = \sum_{i=1}^n [[y_i^1 \neq y_i^2]]$ between configurations (where $[[\cdot]]$ is the Iverson bracket.) Let $\mathcal{B}_k(\mathbf{y})$ denote the k-radius Hamming ball centered at $\mathbf{y}$, *i.e.* $\mathcal{B}(\mathbf{y}) = \{\mathbf{y}' \mid \mathrm{Ham}(\mathbf{y}', \mathbf{y}) \leq k\}$. The previous section constructed one group per label $\ell$. Now, we construct one group $G_\mathbf{y}$ for each configuration $\mathbf{y}$, which is the k-radius Hamming ball centered at $\mathbf{y}$, *i.e.* $G_\mathbf{y} = \mathcal{B}_k(\mathbf{y})$.

**Structured Representation of Marginal Gains.** For this diversity, the marginal gain $d(\mathbf{y}|S)$ becomes a HOP called *cardinality potential* [30]. For count group coverage, this becomes

$$d(\mathbf{y}|S) = \left| \{\mathbf{y}' \mid G_{\mathbf{y}'} \cap (S \cup \mathbf{y}) \neq \emptyset\} \right| - \left| \{\mathbf{y}' \mid G_{\mathbf{y}'} \cap S \neq \emptyset\} \right| \qquad (9a)$$

$$= \left| \bigcup_{\mathbf{y}' \in S \cup \mathbf{y}} \mathcal{B}_k(\mathbf{y}') \right| - \left| \bigcup_{\mathbf{y}' \in S} \mathcal{B}_k(\mathbf{y}') \right| = \left| \mathcal{B}_k(\mathbf{y}) \right| - \left| \mathcal{B}_k(\mathbf{y}) \cap \left[ \bigcup_{\mathbf{y}' \in S} \mathcal{B}_k(\mathbf{y}') \right] \right|, \qquad (9b)$$

*i.e.*, the marginal gain of adding $\mathbf{y}$ is the number of new configurations $\mathbf{y}'$ covered by the Hamming ball centered at $\mathbf{y}$. Since the size of the intersection of $\mathcal{B}_k(\mathbf{y})$ with a union of Hamming balls does not have a straightforward structured representation, we maximize a lower bound on $d(\mathbf{y}|S)$ instead:

$$d(\mathbf{y} \mid S) \geq d_{lb}(\mathbf{y} \mid S) \equiv \left| \mathcal{B}_k(\mathbf{y}) \right| - \sum_{\mathbf{y}' \in S} \left| \mathcal{B}_k(\mathbf{y}) \cap \mathcal{B}_k(\mathbf{y}') \right| \qquad (10)$$

This lower bound $d_{lb}(\mathbf{y}|S)$ overcounts the intersection in Eqn. (9b) by summing the intersections with each $\mathcal{B}_k(\mathbf{y}')$ separately. We can also interpret this lower bound as clipping the series arising from the inclusion-exclusion principle to the first-order terms. Importantly, (10) depends on $\mathbf{y}$ only via its Hamming distance to $\mathbf{y}'$. This is a *cardinality potential* that depends only on the *number* of variables $y_i$ assigned to a particular label. Specifically, ignoring constant terms, the lower bound can be written as a summation of cardinality factors (one for each previous solution $\mathbf{y}' \in S$): $d_{lb}(\mathbf{y}|S) = \sum_{\mathbf{y}' \in S} \theta_{\mathbf{y}'}(\mathbf{y})$, where $\theta_{\mathbf{y}'}(\mathbf{y}) = \frac{b}{|S|} - I_{\mathbf{y}'}(\mathbf{y})$, $b$ is a constant (size of a k-radius Hamming ball), and $I_{\mathbf{y}'}(\mathbf{y})$ is the number of points in the intersection of k-radius Hamming balls centered at $\mathbf{y}'$ and $\mathbf{y}$.

With this approximation, the greedy step means performing MAP inference in a factor graph augmented with cardinality potentials: $\operatorname{argmax}_{\mathbf{y}} r(\mathbf{y}) + \lambda d_{lb}(\mathbf{y}|S)$. This may be solved via message-passing, and all outgoing messages from cardinality factors can be computed in $O(n \log n)$ time [30]. While this algorithm does not offer any approximation guarantees, it performs well in practice. A subtle point to note is that $d_{lb}(\mathbf{y}|S)$ is always decreasing w.r.t. $|S|$ but may become negative due to over-counting. We can fix this by clamping $d_{lb}(\mathbf{y}|S)$ to be greater than 0, but in our experiments this was unnecessary – the greedy algorithm never chose a set where $d_{lb}(\mathbf{y}|S)$ was negative.

**Comparison to DivMBest.** The greedy algorithm for Hamming diversity is similar in spirit to the recent work of Batra et al. [2], who also proposed a greedy algorithm (DivMBest) for finding diverse MAP solutions in graphical models. They did not provide any justification for greedy, and our formulation sheds some light on their work. Similar to our approach, at each greedy step, DivMBest involves maximizing a diversity-augmented score: $\operatorname{argmax}_{\mathbf{y}} r(\mathbf{y}) + \lambda \sum_{\mathbf{y}' \in S} \theta_{\mathbf{y}'}(\mathbf{y})$. However, their diversity function grows *linearly* with the Hamming distance, $\theta_{\mathbf{y}'}(\mathbf{y}) = \operatorname{Ham}(\mathbf{y}', \mathbf{y}) = \sum_{i=1}^{n} [\![y_i' \neq y_i]\!]$. Linear diversity rewards are not robust, and tend to over-reward diversity. Our formulation uses a robust diversity function $\theta_{\mathbf{y}'}(\mathbf{y}) = \frac{b}{|S|} - I_{\mathbf{y}'}(\mathbf{y})$ that saturates as $\mathbf{y}$ moves far away from $\mathbf{y}'$.

In our experiments, we make the saturation behavior smoothly *tunable* via a parameter $\gamma$: $I_{\mathbf{y}'}(\mathbf{y}) = e^{-\gamma \operatorname{Ham}(\mathbf{y}', \mathbf{y})}$. A larger $\gamma$ corresponds to Hamming balls of smaller radius, and can be set to optimize performance on validation data. We found this to work better than directly tuning the radius $k$.

# 4 Experiments

We apply our greedy maximization algorithms to two image segmentation problems: (1) interactive binary segmentation (object cutout) (Section 4.1); (2) category-level object segmentation on the PASCAL VOC 2012 dataset [11] (Section 4.2). We compare all methods by their respective oracle accuracies, i.e. the accuracy of the most accurate segmentation in the set of $M$ diverse segmentations returned by that method. For a small value of $M \approx 5$ to 10, a high oracle accuracy indicates that the algorithm has achieved high recall and has identified a good pool of candidate solutions for further processing in a cascaded pipeline. In both experiments, the label "background" is typically expected to appear somewhere in the image, and thus does not play a role in the label cost/transition diversity functions. Furthermore, in binary segmentation there is only one non-background label. Thus, we report results with Hamming diversity only (label cost and label transition diversities are not applicable). For the multi-class segmentation experiments, we report experiments with all three.

**Baselines.** We compare our proposed methods against DivMBest [2], which greedily produces diverse segmentation by explicitly adding a linear Hamming distance term to the factor graph. Each Hamming term is decomposable along the variables $y_i$ and simply modifies the node potentials $\tilde{\theta}(y_i) = \theta(y_i) + \lambda \sum_{\mathbf{y}' \in S} [\![y_i \neq y_i']\!]$. DivMBest has been shown to outperform techniques such as M-Best-MAP [34, 1], which produce high scoring solutions without a focus on diversity, and sampling-based techniques, which produce diverse solutions without a focus on the relevance term [2]. Hence, we do not include those methods here. We also report results for combining different diversity functions via two operators: ($\otimes$), where we generate the top $\frac{M}{k}$ solutions for each of $k$ diversity functions and then concatenate these lists; and ($\oplus$), where we linearly combine diversity functions (with coefficients chosen by $k$-D grid search) and generate $M$ solutions using the combined diversity.

## 4.1 Interactive segmentation

In interactive foreground-background segmentation, the user provides partial labels via scribbles. One way to minimize interactions is for the system to provide a set of candidate segmentations for the user to choose from. We replicate the experimental setup of [2], who curated 100 images from the PASCAL VOC 2012 dataset, and manually provided scribbles on objects contained in them. For each image, the relevance model $r(\mathbf{y})$ is a 2-label pairwise CRF, with a node term for each

| | Label Cost (LC) | | | | Hamming Ball (HB) | | | | Label Transition (LT) | | |
|---|---|---|---|---|---|---|---|---|---|---|---|
| | MAP | M=5 | M=15 | | MAP | M=5 | M=15 | | MAP | M=5 | M=15 |
| $\min\{1,\cdot\}$ | 42.35 | 45.43 | 45.58 | DivMBest | 43.43 | 51.21 | 52.90 | $\min\{1,\cdot\}$ | 42.35 | 44.26 | 44.78 |
| $\sqrt{(\cdot)}$ | 42.35 | 45.72 | 50.01 | HB | 43.43 | **51.71** | **55.32** | $\sqrt{(\cdot)}$ | 42.35 | 45.43 | 46.21 |
| $\log(1+\cdot)$ | 42.35 | **46.28** | 50.39 | | | | | $\log(1+\cdot)$ | 42.35 | **45.92** | **46.89** |

| | ⊗ Combined Diversity | | | ⊕ Combined Diversity | |
|---|---|---|---|---|---|
| | M=15 | M=16 | | M=15 | |
| HB ⊗ LC ⊗ LT | **56.97** | - | DivMBest ⊕ HB | 55.89 | |
| DivMBest ⊗ HB ⊗ LC ⊗ LT | - | **57.39** | DivMBest ⊕ LC ⊕ LT | 53.47 | |

Table 2: PASCAL VOC 2012 `val` oracle accuracies for different diversity functions.

superpixel in the image and an edge term for each adjacent pair of superpixels. At each superpixel, we extract colour and texture features. We train a Transductive SVM from the partial supervision provided by the user scribbles. The node potentials are derived from the scores of these TSVMs. The edge potentials are contrast-sensitive Potts. Fifty of the images were used for tuning the diversity parameters $\lambda, \gamma$, and the other 50 for reporting oracle accuracies. The 2-label contrast-sensitive Potts model results in a supermodular relevance function $r(\mathbf{y})$, which can be efficiently maximized via graph cuts [20]. The Hamming ball diversity $d_{lb}(\mathbf{y}|S)$ is a collection of cardinality factors, which we optimize with the Cyborg implementation [30].

**Results.** For each of the 50 test images in our dataset we generated the single best $\mathbf{y}^1$ and 5 additional solutions $\{\mathbf{y}^2, \ldots, \mathbf{y}^6\}$ using each method. Table 3 shows the average oracle accuracies for DivMBest, Hamming ball diversity, and their two combinations. We can see that the combinations slightly outperform both approaches.

| | MAP | M=2 | M=6 |
|---|---|---|---|
| DivMBest | 91.57 | 93.16 | 95.02 |
| Hamming Ball | 91.57 | 93.95 | 94.86 |
| DivMBest⊗Hamming Ball | - | - | 95.16 |
| DivMBest⊕Hamming Ball | - | - | 95.14 |

Table 3: Interactive segmentation: oracle pixel accuracies averaged over 50 test images

## 4.2 Category level Segmentation

In category-level object segmentation, we label each pixel with one of 20 object categories or background. We construct a multi-label pairwise CRF on superpixels. Our node potentials are outputs of category-specific regressors trained by [6], and our edge potentials are multi-label Potts. Inference in the presence of diversity terms is performed with the implementations of Delong et al. [9] for label costs, Tarlow et al. [30] for Hamming ball diversity, and Boykov et al. [3] for label transitions.

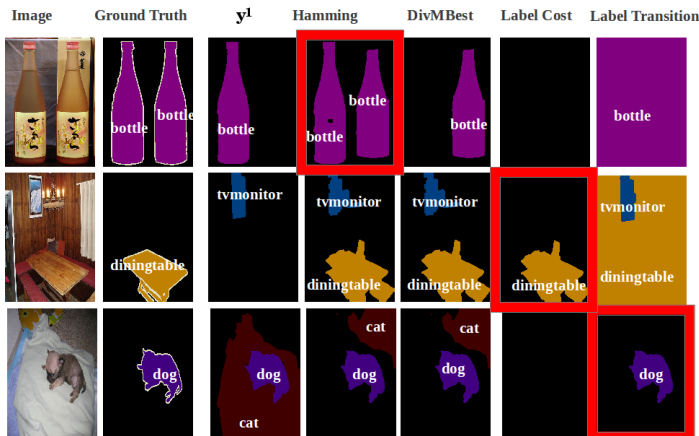

Figure 3: Qualitative Results: each row shows the original image, ground-truth segmentation (GT) from PASCAL, the single-best segmentation $\mathbf{y}^1$, and oracle segmentation from the $M = 15$ segmentations (excluding $\mathbf{y}^1$) for different definitions of diversity. Hamming typically performs the best. In certain situations (row3), label transitions help since the single-best segmentation $\mathbf{y}^1$ included a rare pair of labels (dog-cat boundary).

**Results**. We evaluate all methods on the PASCAL VOC 2012 data [11], consisting of `train`, `val` and `test` partitions with about 1450 images each. We train the regressors of [6] on `train`, and report oracle accuracies of different methods on `val` (we cannot report oracle results on `test` since those annotations are not publicly available). Diversity parameters $(\gamma, \lambda)$ are chosen by performing cross-val on `val`. The standard PASCAL accuracy is the corpus-level intersection-over-union measure, averaged over all categories. For both label cost and transition, we try 3 different concave

functions $h(\cdot) = \min\{1, \cdot\}$, $\sqrt{(\cdot)}$ and $\log(1 + \cdot)$. Table 2 shows the results.[3] Hamming ball diversity performs the best, followed by DivMBest, and label cost/transitions are worse here. We found that while worst on average, label transition diversity helps in an interesting scenario – when the first best segmentation $\mathbf{y}^1$ includes a pair of rare or mutually confusing labels (say dog-cat). Fig. 3 shows an example, and more illustrations are provided in the supplement. In these cases, searching for a different label transition produces a better segmentation. Finally, we note that lists produced with combined diversity significantly outperform any single method (including DivMBest).

## 5   Discussion and Conclusion

In this paper, we study greedy algorithms for maximizing scoring functions that promote diverse sets of combinatorial configurations. This problem arises naturally in domains such as Computer Vision, Natural Language Processing, or Computational Biology, where we want to search for a set of *diverse high-quality* solutions in a structured output space.

The diversity functions we propose are monotone submodular functions by construction. Thus, if $r(\mathbf{y}) + p(\mathbf{y}) \geq 0$ for all $\mathbf{y}$, then the entire scoring function $F$ is monotone submodular. We showed that $r(\mathbf{y})$ can simply be learned to be positive. The greedy algorithm for maximizing monotone submodular functions has proved useful in moderately-sized *unstructured* spaces. To the best of our knowledge, this is the first generalization to exponentially large structured output spaces. In particular, our contribution lies in reducing the greedy augmentation step to inference with structured, efficiently solvable HOPs. This insight makes new connections between submodular optimization and work on inference in graphical models. We now address some questions.

**Can we sample?** One question that may be posed is how random sampling would perform for large ground sets $V$. Unfortunately, the expected value of a random sample of $M$ elements can be much worse than the optimal value $F(S^*)$, especially if $N$ is large. Lemma 1 is proved in the supplement.

**Lemma 1.** *Let $S \subseteq V$ be a sample of size $M$ taken uniformly at random. There exist monotone submodular functions where $\mathbb{E}[F(S)] \leq \frac{M}{N} \max_{|S|=M} F(S)$.*

**Guarantees?** If $F$ is nonnegative, monotone submodular, then using an exact HOP inference algorithm will clearly result in an approximation factor of $1 - 1/e$. But many HOP inference procedures are approximate. Lemma 2 formalizes how approximate inference affects the approximation bounds.

**Lemma 2.** *Let $F \geq 0$ be monotone submodular. If each step of the greedy algorithm uses an approximate marginal gain maximizer $b^{t+1}$ with $F(b^{t+1} \mid S^t) \geq \alpha \max_{a \in V} F(a \mid S^t) - \epsilon_{t+1}$, then $F(S^M) \geq (1 - \frac{1}{e^\alpha}) \max_{|S| \leq M} F(S) - \sum_{i=1}^{M} \epsilon_t$.*

Parts of Lemma 2 have been observed in previous work [13, 29]; we show the combination in the supplement. If $F$ is monotone but not nonnegative, then Lemma 2 can be extended to a relative error bound $\frac{F(S^M) - F_{\min}}{F(S^*) - F_{\min}} \geq (1 - \frac{1}{e^\alpha}) - \frac{\sum_i \epsilon_i}{F(S^*) - F_{\min}}$ that refers to $F_{\min} = \min_S F(S)$ and the optimal solution $S^*$. While stating these results, we add that further additive approximation losses occur if the approximation bound for inference is computed on a shifted or reflected function (positive scores vs positive energies). We pose theoretical improvements as an open question for future work. That said, our experiments convincingly show that the algorithms perform very well in practice, even when there are no guarantees (as with Hamming Ball diversity).

**Generalization.** In addition to the three specific examples in Section 3, our constructions generalize to the broad HOP class of upper-envelope potentials [18]. The details are provided in the supplement.

**Acknowledgements.** We thank Xiao Lin for his help. The majority of this work was done while AP was an intern at Virginia Tech. AP and DB were partially supported by the National Science Foundation under Grant No. IIS-1353694 and IIS-1350553, the Army Research Office YIP Award W911NF-14-1-0180, and the Office of Naval Research Award N00014-14-1-0679, awarded to DB. SJ was supported by gifts from Amazon Web Services, Google, SAP, The Thomas and Stacey Siebel Foundation, Apple, C3Energy, Cisco, Cloudera, EMC, Ericsson, Facebook, GameOnTalis, Guavus, HP, Huawei, Intel, Microsoft, NetApp, Pivotal, Splunk, Virdata, VMware, WANdisco, and Yahoo!.

## Footnotes

[1]The accuracy of the most accurate segmentation in the set.

[2]Strictly speaking, this condition is sufficient but not necessary. We only need nonnegative marginal gains.

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
