[Supplementary Material]

# Supplementary Material:
# Submodular meets Structured: Finding Diverse Subsets in Exponentially-Large Structured Item Sets

**Adarsh Prasad**
UT Austin
adarsh@cs.utexas.edu

**Stefanie Jegelka**
UC Berkeley
stefje@eecs.berkeley.edu

**Dhruv Batra**
Virginia Tech
dbatra@vt.edu

## 1 Structured SVMs with nonnegativity constraint

In this section, we will show that SSVMs have no natural origin, and that parameters learnt with non-negativity constraints achieve exactly the same hinge loss as achieved without the nonnegativity constraint.

For a set of $\ell$ training instances $(\boldsymbol{x}^n, \mathbf{y}^n) \in \mathcal{X} \times \mathcal{Y}, n = 1, \ldots, \ell$ from a sample space $\mathcal{X}$ and label space $\mathcal{Y}$, the structured SVM minimizes the following regularized risk function.

$$\min_{\boldsymbol{w}} \quad \|\boldsymbol{w}\|^2 + C \sum_{n=1}^{\ell} \max_{\mathbf{y} \in \mathcal{Y}} \left( \Delta(\mathbf{y}^n, \mathbf{y}) + \boldsymbol{w}'\Psi(\boldsymbol{x}^n, \mathbf{y}) - \boldsymbol{w}'\Psi(\boldsymbol{x}^n, \mathbf{y}^n) \right) \tag{1}$$

The function $\Delta : \mathcal{Y} \times \mathcal{Y} \to \mathbb{R}_+$ measures a distance in label space and is an arbitrary function satisfying $\Delta(\mathbf{y}, \mathbf{y}') \geq 0$ and $\Delta(\mathbf{y}, \mathbf{y}) = 0 \;\; \forall \mathbf{y}, \mathbf{y}' \in \mathcal{Y}$. The function $\Psi : \mathcal{X} \times \mathcal{Y} \to \mathbb{R}^d$ is a feature function, extracting some feature vector from a given sample and label.

Because the regularized risk function above is non-differentiable, it is often reformulated in terms of a quadratic program by introducing one slack variable $\xi_n$ for each sample, each representing the value of the maximum. The standard structured SVM primal formulation is given as follows.

$$\begin{aligned}
\min_{\boldsymbol{w}, \boldsymbol{\xi}} \quad & \|\boldsymbol{w}\|^2 + C \sum_{n=1}^{\ell} \xi_n \\
\text{s.t.} \quad & \boldsymbol{w}'\Psi(\boldsymbol{x}^n, \mathbf{y}^n) - \boldsymbol{w}'\Psi(\boldsymbol{x}^n, \mathbf{y}) + \xi_n \geq \Delta(\mathbf{y}^n, \mathbf{y}), \qquad n = 1, \ldots, \ell, \quad \forall \mathbf{y} \in \mathcal{Y}
\end{aligned} \tag{2}$$

We will refer to the primal objective in Equation 2 as $\boldsymbol{P}_1$ and the optimal solution as $\boldsymbol{w}_1, \boldsymbol{\xi}_1$.

Next, consider the following augmented formulation, where $\hat{\boldsymbol{w}} = \begin{bmatrix} w \\ b \end{bmatrix}, \hat{\Psi}(\boldsymbol{x}^n, \mathbf{y}^n) = \begin{bmatrix} \Psi(\boldsymbol{x}^n, \mathbf{y}^n) \\ 1 \end{bmatrix}$:

$$\begin{aligned}
\min_{\hat{\boldsymbol{w}}, \boldsymbol{\xi}} \quad & \|\boldsymbol{w}\|^2 + C \sum_{n=1}^{\ell} \xi_n \\
\text{s.t.} \quad & \hat{\boldsymbol{w}}'\hat{\Psi}(\boldsymbol{x}^n, \mathbf{y}^n) - \hat{\boldsymbol{w}}'\hat{\Psi}(\boldsymbol{x}^n, \mathbf{y}) + \xi_n \geq \Delta(\mathbf{y}^n, \mathbf{y}), \qquad n = 1, \ldots, \ell, \quad \forall \mathbf{y} \in \mathcal{Y}
\end{aligned} \tag{3}$$

We will refer to the primal objective in Equation 3 as $\boldsymbol{P}_2$ and the optimal solution as $\hat{\boldsymbol{w}}_2 = \begin{bmatrix} w_2 \\ b_2 \end{bmatrix}, \boldsymbol{\xi}_2$.

*Note: We do not regularize b.*

**Claim 1.** $\boldsymbol{P}_1(\boldsymbol{w}_1, \boldsymbol{\xi}_1) = \boldsymbol{P}_2(\hat{\boldsymbol{w}}_2, \boldsymbol{\xi}_2)$
**Proof.** Adding a bias feature (b) doesn't affect the objective function and every constraint is invariant of it. Hence, the two problems are equivalent.

Next, consider the formulation, where we extend $\boldsymbol{P}_2$ to enforce non-negativity of scores.

$$
\begin{aligned}
\min_{\hat{\boldsymbol{w}}, \boldsymbol{\xi}} \quad & \|\boldsymbol{w}\|^2 + C \sum_{n=1}^{\ell} \xi_n \\
\text{s.t.} \quad & \hat{\boldsymbol{w}}' \hat{\Psi}(\boldsymbol{x}^n, \mathbf{y}^n) - \hat{\boldsymbol{w}}' \hat{\Psi}(\boldsymbol{x}^n, \mathbf{y}) + \xi_n \geq \Delta(\mathbf{y}^n, \mathbf{y}), \qquad n = 1, \ldots, \ell, \quad \forall \mathbf{y} \in \mathcal{Y} \\
& \hat{\boldsymbol{w}}' \hat{\Psi}(\boldsymbol{x}^n, \mathbf{y}) \geq 0 \qquad n = 1, \ldots, \ell, \quad \forall \mathbf{y} \in \mathcal{Y}
\end{aligned}
\tag{4}
$$

We will refer to the primal objective in Equation 4 as $\boldsymbol{P}_3$ and the optimal solution as $\hat{\boldsymbol{w}}_3 = \begin{bmatrix} w_3 \\ b_3 \end{bmatrix}, \boldsymbol{\xi}_3$.

**Claim 2.** $\boldsymbol{P}_2(\hat{\boldsymbol{w}}_2, \boldsymbol{\xi}_2) = \boldsymbol{P}_3(\hat{\boldsymbol{w}}_3, \boldsymbol{\xi}_3)$

**Proof.** It is easy to see that $\boldsymbol{P}_2(\hat{\boldsymbol{w}}_2, \boldsymbol{\xi}_2) \leq \boldsymbol{P}_3(\hat{\boldsymbol{w}}_3, \boldsymbol{\xi}_3)$ as $(\hat{\boldsymbol{w}}_2, \boldsymbol{\xi}_2)$ is the optimal solution for $\boldsymbol{P}_2$ and $(\hat{\boldsymbol{w}}_3, \boldsymbol{\xi}_3)$ is also a feasible solution for $\boldsymbol{P}_2$.

Consider the vector $\hat{\boldsymbol{w}}_2^* = \begin{bmatrix} w_2 \\ -\min_n \min_{\mathbf{y}} w_2' \Psi(\boldsymbol{x}^n, \mathbf{y}) \end{bmatrix}$, Since the bias term doesn't occur in the objective function, hence $\boldsymbol{P}_3(\hat{\boldsymbol{w}}_2^*, \boldsymbol{\xi}_2) = \boldsymbol{P}_2(\hat{\boldsymbol{w}}_2, \boldsymbol{\xi}_2)$.

Also, $(\hat{\boldsymbol{w}}_2^*, \boldsymbol{\xi}_2)$ is a feasible solution to $\boldsymbol{P}_3$, hence, $\boldsymbol{P}_3(\hat{\boldsymbol{w}}_3, \boldsymbol{\xi}_3) \leq \boldsymbol{P}_3(\hat{\boldsymbol{w}}_2^*, \boldsymbol{\xi}_2) = \boldsymbol{P}_2(\hat{\boldsymbol{w}}_2, \boldsymbol{\xi}_2)$. Therefore, $\boldsymbol{P}_2(\hat{\boldsymbol{w}}_2, \boldsymbol{\xi}_2) = \boldsymbol{P}_3(\hat{\boldsymbol{w}}_3, \boldsymbol{\xi}_3)$.

Therefore, even after adding the nonnegativity constraints, the solutions achieve the same values for the *regularised risk function* and hence are expected to have the same *generalization* guarantees. In practice, the non-negativity constraints can be added in a cutting-plane procedure via a MAP call.

## 2 Label Transitions

**Groups and Motivating Scenario.** In this section, we generalize the label cost diversity function to reward not just the presence of certain labels, but the presence of certain *label transitions*. For instance, if the highest scoring segmentation contains a "cow" on "grass", this diversity function will reward other segmentations for containing novel label transitions, such as "sheep-grass" or "cow-ground" or "sheep-sky". Formally, we define one group $G_{\ell, \ell'}$ per label pair $\ell, \ell'$, and an item $a$ belongs to $G_{\ell, \ell'}$ if $\mathbf{y} = \phi(a)$ contains two adjacent variables $y_i, y_j$ with labels $y_i = \ell, y_j = \ell'$.

**Structured Representation of Marginal Gains.** For diversity of label transitions, the marginal gain $D(a \mid S)$ becomes a HOP called *cooperative cuts* [1]. Let $\mathrm{Cut}_{\mathbf{y}}(\ell, \ell') = \{(y_i, y_j) \in \mathcal{E} \mid y_i = \ell, y_j = \ell'\}$ be the cut set for a specific label transition $(\ell, \ell')$. Further, let $\#\mathrm{Cut}_S(\ell, \ell')$ count the number of items $a \in S$ that contain at least one $(\ell, \ell')$ label transition: $\#\mathrm{Cut}_S(\ell, \ell') = |\{a \in S \mid \mathrm{Cut}_{\phi(a)}(\ell, \ell') \neq \emptyset\}| = |S \cap G_{\ell, \ell'}|$. The marginal gains for this diversity function are:

$$
d(\mathbf{y} \mid S) = D(\phi^{-1}(\mathbf{y}) \mid S) \tag{5a}
$$

$$
= \sum_{\ell, \ell'} h(\#\mathrm{Cut}_{S \cup \phi^{-1}(\mathbf{y})}(\ell, \ell')) - h(\#\mathrm{Cut}_S(\ell, \ell')) \tag{5b}
$$

$$
= \sum_{\substack{\ell, \ell' \\ \mathrm{Cut}_{\mathbf{y}}(\ell, \ell') \neq \emptyset}} h(1 + \#\mathrm{Cut}_S(\ell, \ell')) - h(\#\mathrm{Cut}_S(\ell, \ell')) \tag{5c}
$$

Similar to single label groups, the gain for a label pair $(\ell, \ell')$ decreases as $\#\mathrm{Cut}_S(\ell, \ell')$ grows. Thus, $d(\mathbf{y} \mid S)$ rewards the presence of pair $(\ell, \ell')$ by an amount proportional to how rare it is in the segmentations in $S$. Analogously to the label costs, the parsimony factor in this setting is $p(\mathbf{y}) = \sum_{\mathrm{Cut}_{\mathbf{y}}(\ell, \ell') \neq \emptyset} c(\ell, \ell')$ encouraging each individual $\mathbf{y}$ to have a small number of label transitions. Specifically, when using a count coverage and parsimony term with $c(\ell, \ell') = -1$, we eventually maximize

$$
r(\mathbf{y}) + p(\mathbf{y}) + d(\mathbf{y}) \tag{6a}
$$

$$
= r(\mathbf{y}) - \sum_{\ell, \ell': S \in G_{\ell, \ell'}} \min\{\#\mathrm{Cut}_{\mathbf{y}}(\ell, \ell'), 1\}, \tag{6b}
$$

Figure 1: Color map for reading VOC segmentation results.

which is a supermodular function on the set of cut edges for each label transition. Thus, the multi-label cooperative cut inference algorithm by Kohli *et al*. [2] applies. The construction for general $h$ looks similar, with a degrading cost in front of the min.

## 3   Experiments

For the sake of completeness and to show the difference in sets of solutions generated by different diversity functions, we show sample sets of solutions generated for a given image (Fig. 2, 3 and 4). These results help in understanding the behavior of different diversity functions.

Figure 2: Sets of solutions generated with different diversity functions.

Image&emsp;&emsp;Ground Truth

|  | $y^1$ | $y^2$ | $y^3$ | $y^4$ | $y^5$ |
|---|---|---|---|---|---|
| **Hamming** | 50.77% | 25.56% | 19.01% | 92.51% | 4.78% |
| **DivMbest** | 50.77% | 20.63% | 50.77% | 19.36% | 49.40% |
| **Label Cost** | 50.77% | 12.47% | 0.00% | 5.05% | 23.31% |
| **Label Transition** | 50.77% | 0.00% | 0.00% | 23.31% | 23.31% |

Image&emsp;&emsp;Ground Truth

|  | $y^1$ | $y^2$ | $y^3$ | $y^4$ | $y^5$ |
|---|---|---|---|---|---|
| **Hamming** | 34.23% | 28.97% | 38.74% | 47.01% | 31.00% |
| **DivMbest** | 34.23% | 30.03% | 44.72% | 5.28% | 44.72% |
| **Label Cost** | 34.23% | 0.00% | 0.00% | 0.00% | 5.06% |
| **Label Transition** | 34.23% | 0.00% | 0.00% | 0.00% | 0.00% |

Figure 3: Sets of solutions generated with different diversity functions.

Figure 4: Sets of solutions generated with different diversity functions.

# 4 Proof of Lemma 1

**Lemma 1.** *Let $S$ be a sample of size $M$ taken uniformly at random. There exist monotone submodular functions where $\mathbb{E}[F(S)] \leq (M/N + \epsilon/M) \max_{|S| \leq M} F(S)$ for any $\epsilon \geq 0$.*

The bound in the main paper follows with $\epsilon = 0$.

*Proof.* To prove this statement, we consider a specific worst-case function. Let $R \subseteq V$ be a fixed set of size $M$, and let

$$F(S) = |S \cap R| + \epsilon \min\{|S \setminus R|, 1\}. \tag{7}$$

The function $F$ is obviously nondecreasing and it is also submodular. For this function, the cardinality-constrained optimum is

$$\max_{|S| \leq M} F(S) = F(R) = M. \tag{8}$$

The expected value of an $M$-sized sample is the expectation of a hypergeometric distribution plus a correction taking into account that every set except $R$ will have value at least 1 (using the second part of $F$):

$$\mathbb{E}_S[F(S)] = \binom{N}{M}^{-1} \left( \sum_{S \subseteq V, |S|=M} |S \cap R| + \binom{N}{M} \epsilon - \epsilon \right) \tag{9}$$

$$= \binom{N}{M}^{-1} \left( \sum_{r=1}^{M} \binom{M}{r} \binom{N-M}{M-r} r - \epsilon \right) + \epsilon \tag{10}$$

$$= \frac{M^2}{N} + \epsilon - \binom{N}{M}^{-1} \epsilon \tag{11}$$

$$< (M/N + \epsilon/M)F(R). \tag{12}$$

$\square$

This is also a fairly tight bound: if we sample each element with probability $M/N$, then, using Lemma 2.2 in [3] and the monotonicity of $F$, it holds that $\mathbb{E}_S[F(S)] \geq \frac{M}{N} F(V) \geq \frac{M}{N} F(S^*)$.

## 5   Proof of Lemma 2

**Lemma 2.** *Let $F \geq 0$ be monotone submodular. If each step of the greedy algorithm uses an approximate gain maximizer $b^{i+1}$ with $F(b^{i+1} \mid S^i) \geq \alpha \max_{a \in V} F(a \mid S^i) - \epsilon^{i+1}$, then*

$$F(S^M) \geq (1 - \frac{1}{e^\alpha}) \max_{|S| \leq M} F(S) - \sum_{i=1}^{M} \epsilon^i.$$

To prove Lemma 2, we employ a helpful intermediate observation. We will denote the optimal solution by $S^* \in \arg\max_{|S| \leq M} F(S)$.

**Lemma 3.** *If $F(b^{i+1} \mid S^i) \geq \alpha \max_{a \in V} F(a \mid S^i) - \epsilon^{i+1}$, then*

$$F(b^{i+1} \mid S^i) \geq \frac{\alpha}{M}(F(S^*) - F(S^i)) - \epsilon^{i+1}.$$

*Proof. (Lemma 3).* Define $T^i = S^* \setminus S^i$ to be the set of all elements that are in $S^*$ but have not yet been selected. Order the elements in $T^i$ in an arbitrary order as $t^1, \ldots t^h$ (note that $h \leq M$ because

$|S^*| \leq M$. By monotonicity of $F$ and the fact that $S^* \subseteq S^i \cup T^i$, it holds that

$$F(S^*) - F(S^i) \leq F(T^i \cup S^i) - F(S^i) \tag{13}$$

$$= \sum_{j=1}^{h} F(t^j \mid S^i \cup \{t^1, \dots, t^{j-1}\}) \tag{14}$$

$$\leq \sum_{j=1}^{h} F(t^j \mid S^i) \tag{15}$$

$$\leq M \max_{a \in V \setminus S^i} F(a \mid S^i) \tag{16}$$

$$\leq \frac{M}{\alpha}(F(b^{i+1} \mid S^i) + \epsilon^{i+1}) \tag{17}$$

In Equation (15), we use diminishing marginal returns. In the end, we use the assumption of the lemma, which implies that

$$\max_a F(a \mid S) \leq \frac{1}{\alpha}(F(b^{i+1} \mid S^i) + \epsilon^{i+1}). \tag{18}$$

Rearranging yields the result of the lemma. □

Now we are equipped to prove Lemma 2.

*Proof.* Lemma 3 implies that

$$F(b^{i+1} \mid S^i) = F(S^{i+1}) - F(S^i) \tag{19}$$

$$\geq \frac{\alpha}{M}\big(F(S^*) - F(S^i)\big) - \epsilon^{i+1}. \tag{20}$$

We rearrange this to

$$F(S^*) - F(S^{i+1}) \tag{21}$$

$$\leq (1 - \frac{\alpha}{M})F(S^*) - (1 - \frac{\alpha}{M})F(S^i) + \epsilon^{i+1} \tag{22}$$

$$= (1 - \frac{\alpha}{M})(F(S^*) - F(S^i)) + \epsilon^{i+1} \tag{23}$$

$$\leq (1 - \frac{\alpha}{M})^{i+1}(F(S^*) - F(S^0)) + \sum_{j=1}^{i} \big(1 - \frac{\alpha}{M}\big)^{i-j}\epsilon^j \tag{24}$$

$$\leq (1 - \frac{\alpha}{M})^{i+1}(F(S^*) - F(S^0)) + \sum_{j=1}^{i} \epsilon^j \tag{25}$$

With $F(S^0) = F(\emptyset) = 0$ we rearrange to

$$F(S^M) \geq (1 - (1 - \frac{\alpha}{M})^M)F(S^*) - \sum_{j=1}^{M} \epsilon^j \tag{26}$$

$$\geq (1 - e^{-\alpha})F(S^*) - \sum_{j=1}^{M} \epsilon^j. \tag{27}$$

□

# 6   Relative Error

If we have a monotone but not nonnegative function, we may shift the function and obtain a relative bound:

**Lemma 4.** *Let $F$ be an arbitrary monotone submodular function, and let $F_{\min} = \min_{S \subseteq V} F(S)$. We can define a new, shifted monotone non-negative version of $F$ as $F^+(S) \triangleq F(S) - F_{\min}$. If we apply the greedy algorithm to $F^+$, we obtain a solution $\widehat{S}$ that satisfies $F^+(\widehat{S}) \geq \alpha F^+(S^*)$, then the solution $\widehat{S}$ has a bounded relative approximation error:*

$$\frac{F(\widehat{S}) - F_{\min}}{F(S^*) - F_{\min}} \geq \alpha.$$

*Proof.* By a proof analogous to that of Lemma 2, we get, for $\alpha = (1 - \frac{1}{e})$,

$$F^+(\widehat{S}) \geq \alpha F^+(S^*) \tag{28}$$

$$\Leftrightarrow \quad F(\widehat{S}) - F_{\min} \geq \alpha(F(S^*) - F_{\min}) \tag{29}$$

$$\Leftrightarrow \quad \frac{F(\widehat{S}) - F_{\min}}{F(S^*) - F_{\min}} \geq \alpha = 1 - \frac{1}{e}. \tag{30}$$

$\square$

# 7 Generalization: Upper Envelope Potentials

In addition to the three specific examples given in the main document (Section 4.1, 4.1, 4.2), we can also generalize these constructions to a broad class of HOPs called upper envelope potentials [4].

Let $G_1, G_2, \ldots, G_q, \ldots, G_b$ be disjoint groups where $b$ is polynomial in the size of the number of base variables $(n)$. We consider the group count diversity. At iteration $t$ in the greedy algorithm, assume without loss of generality that we have covered $G_1, G_2, \ldots, G_k$. Then for the $(t+1)^{th}$ iteration, the marginal gain of $\mathbf{y}$ is:

$$d(\mathbf{y} \mid S^t) = \begin{cases} 0 & \text{if } \mathbf{y} \in \{G_1, .., G_k\} \\ 1 & \text{otherwise.} \end{cases} \tag{31}$$

We can now express $d(\mathbf{y} \mid S^t)$ as an upper-envelope potential, *i.e.* $d(\mathbf{y} \mid S^t) \equiv \max_q \nu^q(\mathbf{y})$ where:

$$\nu^q(\mathbf{y}) = \mu^q + \sum_{i \in [n]} \sum_{\ell \in L} w_{i\ell}^q \delta_i(\ell) \tag{32}$$

where $\delta_i(\ell)$ returns 1 if $y_i = \ell$.

Each linear function $\nu^q(\mathbf{y})$ encodes two pieces of information:

1. $\mu^q$ encodes if $G_q$ is uncovered at the end of the $t^{th}$ step, *i.e.*

$$\mu^q = \begin{cases} 0 & \text{if } q \in \{1, .., k\} \\ 1 & \text{otherwise.} \end{cases} \tag{33}$$

2. The second part of (32) indicates whether or not $\mathbf{y}$ lies in the group $G_q$ (defined below).

For general groups, it may not be possible to have linear encodings for membership. By construction, we describe one practical example:

**Region Consistency Diversity.** Consider a region $C$ in the image whose superpixels $\mathbf{y}_C$ we want to bias towards a homogenous or uniform labeling. Thus, when we search for diverse labelings, we want to encourage the entire set of superpixels $\mathbf{y}_c$ to change together. In this case, each group $G_\ell$ corresponds to the set of segmentations that assign label $\ell$ to the region $\mathbf{y}_c$. For such a diversity function, we define $w_{i\ell}^q$ as:

$$w_{i\ell}^q = \begin{cases} 0 & \text{if } i \in [C] \text{ and } \ell = q \\ -\infty & \text{otherwise.} \end{cases} \tag{34}$$

With this definition of $w_{i\ell}^q$, the second part of (32) indicates whether or not $\mathbf{y}$ lies in the group $G_q$

It is known [5, 4] that maximizing any upper-envelope HOP can be reduced to the maximization of a pairwise function with the addition of an auxiliary switching variable $x$ that takes values from the index set of $q$ (in this case $[L]$).

$$\max_{\mathbf{y}_c} d(\mathbf{y} \mid S^t) = \max_{\mathbf{y}_c, x} (\phi_x(q) + \sum_{i \in V} \phi_{xi}(q, y_i)) \tag{35}$$

where $\phi_x(q) = \mu^q$ and $\phi_{xi}(q, y_i) = w_{iy_i}^q$. This pairwise function can be maximised using standard message passing algorithms such as TRW and BP. However, for some cases, such as the region consistency diversity defined above, the pairwise function is supermodular, and graph cuts can be used.