[Reviews · NeurIPS 2014]

Submitted by Assigned_Reviewer_3

This paper addresses the problem of large solution spaces in structured prediction problems. They propose greedy algorithms for finding solutions in structured-output spaces. In structured prediction problems, sometimes there is a need for not a single MAP-state but a set of good predictions. These are particularly useful in pipeline models where a MAP state at each stage is not necessarily the best for the next stage. They make use of submodularity property to construct a greedy algorithm to choose the item which has largest marginal gain with an approximation. But, choosing a set from a ground set |V| using greedy algorithm if V is exponential is not feasible. In this paper, they explore conditions that make this feasible. They use structured representations to reduce greedy search to a MAP-inference problem. This idea of connecting the greedy search for solution in exponentially large spaces to MAP-inference in a structured output space is the key idea behind this work.
In this work, they use a single trained model to generate multiple solutions, rather than re-training the model, which is computationally expensive.

The paper makes an interesting connection between submodularity and inference in graphical models. The extension to exponentially large output spaces that are structural in nature is a solid and novel contribution. The experiments bring out the significance of having set of good solutions rather than having a single MAP solution. The third example in figure 3 (dog-cat) brings out the problem in using a single label transition and brings out the importance of searching for different label-transitions.

In the experiments they have considered pair-wise CRF as the structured representation. I am curious to know for how this scales for other complex structured spaces for which only approximate inference is possible.
Summary: The contribution made by the paper in drawing a connection between submodularity and inference in graphical models is interesting. The experiments support their claim. They have considered only a particular class of graphical models in their inference and have limited themselves to pair-wise graph structures. More experiments in this regard can make this work more significant.

Submitted by Assigned_Reviewer_14

The paper proposes to greedily find a series of M diverse solutions to a structured output problem. First, the structured output prediction problem is expressed in terms of MAP inference in a factor graph. Next, this graph is augmented by a higher-order factor that encourages diversity from previously found solutions. Repeated MAP inference in this augmented factor graph (with iteratively updated higher-order potentials) then gives a series of M diverse solutions. Different task-specific user-defined diversity measures lead to inference problems that are a special case of either label costs, co-operative cuts or cardinality potentials inference. The experiments show similar performance to Diverse M-Best [1, cited] on two datasets for one proposed diversity measure whereas the other proposed diversity measures perform significantly worse than [1] in general, but have some advantages on single instances of the dataset.

Quality:
The paper is well written (apart from some minor notational issues raised below) and the structure is mostly clear. It offers some theoretical insights in the DivMBest method and extends that model to higher-order diversity potentials. The paper contains most relevant proofs.

The proposed diversity measures, however, seem to lack some desirable properties:
* The diversity measures in Eqs. (5) and (6) are invariant to label flips. For instance, if in one segmentation, two objects are labeled as cat and dog, then there is no diversity-related reward for flipping those two labels, i.e. labeling the objects as dog and cat, respectively.
* Only new labels are rewarded, structure (location and size of the superpixels,...) is not taken into account, e.g. labeling a different object as dog, or increasing the size of the dog object.
* The greedy fashion of the algorithm has drawbacks when the diversity measure from Eq. (5) is utilized: For instance, if in Fig. 2a the MAP solution y_1 yields a segmentation containing three (super)pixels with a dog, person and horse, then there is no further marginal gain for diversity. Also, the general diversity in Eq. (6) will favor segmentations with many distinct labels. Diverse solutions may, however, also contain fewer distinct labels.
* The parsimony factor raises some concerns:
i) The parsimony factor should naturally be included into the diversity term, not the relevance term (l. 227) since the learned weights in the relevance term should be independent of the diversity term. If a diversity term without bias towards more labels is designed, the parsimony factor would be unnecessary. In other words, if the MAP solution of the factor graph without diversity term is to be found (i.e. minimization of the relevance term), the parsimony factor should not be included.
ii) It is currently chosen in a rather heuristic manner (e.g. how to choose/learn c(l) in line 248?).
iii) It is unclear whether/how this term is weighted against the relevance AND/OR diversity term: In l. 252 it is assigned the same weight as the diversity term, whereas this lambda is missing in l. 396.
iv) Both background-only segmentations and a segmentation containing only new labels are rewarded by zero (using l. 252). Hence, many segmentations will only consist of background, such as in Fig. 3.
v) Another major concern is that the sum in l. 252 may become negative, and consequently the requirements for the objective in Eq. (2) are violated.

Furthermore, the paper does not elaborate on how the maximization in Eq. (2) is done over V\S^(t-1). If the maximization is done over V instead, it is not clear how it can be ensured that an optimal solution a^t is chosen which is not already contained in S^(t-1).

The concerns above are mirrored by the experiments which perform significantly worse than the competitive method [1]. The Hamming balls potentials address some of these issues and achieve competitive results, but approximation errors from maximizing the loose lower bound (eq. 10) are included. Why is it not possible to include the exact marginal gain into the higher-order potential? Moreover, the Hamming ball section would profit from some reformulations and derivations, since this -- according to the experimental results -- seems to be the only useful diversity measure proposed in this paper.

The experimental section would benefit from a more thorough analysis:
* It would be interesting to see how robust the results are with different diversity parameters.
* A combination of DivMBest and the proposed method, i.e. including the unary potentials from DivMBest AND the higher-order potentials, may perform much better.
* It is not stated how the parameters are chosen for the experiment in 5.2, how the parsimony factor weight is chosen, and how the Hamming ball radius k chosen.
* It is not clear why the MAP solutions in Tab. 2 are different.
* Why is only recall reported?

While the authors claim "significantly higher oracle accuracy" (l. 94), according to the Results Section the method performs significantly worse than DivMBest in all cases except for the Hamming ball measure (which they state is rather similar to DivMBest), where results are comparable.

Clarity:
* Fig. 2a (not referenced in the text) shows 6 examples which seem to be diverse segmentations. With the proposed count diversity, however, the greedy algorithm will only reward diversity for the first 1-3 segmentations (dependent on the order). Hence, this example should only include 1-3 different segmentations.
* The notation in Eq. (9b), | B | and | union(B) |, where B are balls, is unclear.
* The derivation of the equation in l. 287 is unclear and so is Fig. 2b: In Fig. 2b, some balls seem to have b=9, some b=7,...; |S|=4. It is not intuitively clear, why b/|S| is used. This makes it difficult to understand the exact connection to DivMBest [1]. The supplementary material should be extended appropriately.
* I_y in ll. 289 and 307 does not seem to be the same.
* Suppl. Fig 4 bottom is confusing: How can y_2 (Hamming) be black completely and y_3 contain a label? y_2 shouldn't have changed the objective, right?
* l. 98: [14] do not learn a MIXTURE of M models
* It is not mentioned in Sec. 3 that r(y) is required to be submodular.
* The relevance function must be non-negative, as stated in l. 160 and enforced by the constraint in Suppl. l. 58. I assume the constraint is added in a cutting-plane fashion, since it involves all y \in Y. In practice, this constraint may be violated for some y \in Y? What is the procedure then?

Originality and Significance:
The derivation of diverse M-best solutions using set functions and maximizing their marginal gains is novel and sheds new light on DivMBest [1] for this specific model class. Extending [1] to higher-order potentials seems promising, but experimental results show only incremental and partial improvement compared to [1].

Minor comments:
Equations 4, 5, 8, 9a: Instead of { G_i | ... } these sets should be over indices of groups { i | ... } rather than over one group G_i
157: r(y) non-negative only ensures that r(.) is monotone.
Fig. 2 is not referenced in the text
19: is a challenging in structured prediction problems -> is a challenging task in structured prediction problems
41: solve [...] problems that often fairly ambiguous -> solve [...] problems that often are fairly ambiguous
81: where [k] is... -> leave out, since it is defined by [L] = {1,...,L} already
143: there is broad class -> there is a broad class
146: use structured a relevance function -> use a structured relevance function
156: and a feature vectors -> and feature vectors
157: It must be stressed that r(.) is only monotone if r(.) is linear and all summands are non-negative.
239: diverity -> diversity
245: missing brackets around the difference
Supp. 66 and below: w* -> \hat w*
266: leave out "large"
289: I(y') -> I(y)
292: presentend -> presented

-----------

I have increased the quality score by 2, assuming the authors will make all promised additions to the camera-ready version.
Summary: This work generalizes DivMBest [1] to higher-order potentials with restrictions on the model class, but performs worse than DivMBest in most cases. This may be due to flaws in the diversity measures as discussed above. Furthermore, the connection to DivMBest lacks some important details.

Submitted by Assigned_Reviewer_19

In this paper, the authors try to connect submodular maximization and graphical models with high-order potentials. They show how to construct submodular diversity functions and structured representation of marginal gains with two examples. In the experiments it shows comparable results compared to DivMBest. The paper provides a good insight of submodular functions and graphical models with HOP. One weakness of this paper is its experiments. Given the related work mentioned in the introduction and [6] mentioned in section 4.1 and the fact that in Table 3, the results are sometimes better and sometimes worse, it is hard to see the effectiveness of the introduced framework.
Summary: In this paper, the authors try to connect submodular maximization and graphical models with high-order potentials. The connect is novel and sound and it will be better to compare with more related work mentioned in the introduction and section 4.1.
Author Feedback
Author rebuttal: We thank the reviewers for their time and feedback.
All reviewers found the proposed connections between diversity/submodularity in structured outputs and inference in graphical models with Higher Order Potentials (HOPs) interesting and novel.

R1 expressed some concerns that we address below.
We hope this new information will help shape the opinion of this paper. Due to limited space, we can only respond to the major comments, but are happy to incorporate all feedback.

We denote LC=LabelCost, LT=LabelTransition, HB=HammingBall, and DmB=DivMBest.

1. R1: "Lack of desirable properties in diversity functions"
We fully agree with the intuitions of R1 regarding (5),(6). However, we would like to stress that there isn't one single gold-standard diversity function that works in all cases.
As we illustrate in the paper, LC and LT each work well in some narrow regimes, but not across the entire dataset on average.
Our goal in developing multiple diversity functions (LC, LT, HB, UpperEnvelope, RegionConsistency) was to show that several task-specific diversity functions could be formulated in a formal justified way in our submodular maximization framework, and to illustrate that greedy augmentation sub-routines for these diversity functions have already been developed.
We will continue to develop better diversity functions incorporating the suggestions by R1. Finally, please note the improved results with combined diversity functions in the next point.

2. R1/R2: "Worse performance than DmB in experiments"
First, note that HB diversity already outperforms DmB in Pascal segmentation (55.32 vs 52.50). LC and LT do not. However, no single diversity function is perfect. Thus, as suggested by R1, we combined diversity functions.
There are two ways of combining:
(A) Generate top M/k from k diversity functions, concatenate these lists to form the full list.
(B) Linear combination of diversity functions (with coefficients chosen by k-D grid search), and M solutions under combined diversity.

Pascal VOC Results:
(A3) Top5 solutions from LC, LT, and HB to form a list of 15 solutions (improv=improvement over DmB alone):
oracle acc 56.97%, improv=4.5% (DmB: 52.50%).
(A4) Top4 solutions from LC, LT, HB, and DmB:
oracle acc 57.39%, improv=4.9% (DmB: 52.50).

(B) Linear combination diversity (improv = improvement over DmB):
DmB + LC: 53.33; improv= 0.8%
DmB + LT: 53.01; improv= 0.5%
DmB + LC + LT: 53.47; improv= 1%
DmB + HB: 55.89; improv= 3.4%
We are unable to report combined numbers for all four in this rebuttal due to logistical problems (HB implementation uses code from Tarlow and LC from Delong). We will report these for the camera ready.
Overall, we can see that combining our proposed diversity functions outperforms previous state-of-art (DmB) by 4.5% (not including DmB) and 4.9% (including DmB).
These results further strengthen our point that while some diversity functions may not do well individually, they do well as a group, and hence in pipeline-based systems, it is beneficial to use different diversities. These results (on pascalseg & binaryseg) will be added to the final version.

3. R1: maximization in Eq.(2) over V\S^(t-1)

This is a great subtle point, thank you for bringing this up. While the manuscript is written with a set-theoretic notation, we are essentially dealing with submodular /sequence/ prediction. The generalization from sets to sequences (or lists) allows reasoning about an ordering of the items chosen and repeated entries in the list. More details about list prediction can be found in Streeter & Golovin NIPS09, Dey et al. RSS12, Ross et al. ICML13.
Essentially this means that the maximization in Eq. (2) is done over V (there may be duplicates in our lists; in practice this is very rare). We will clarify this in the final version.

4. Other clarifications:
- Parsimony Terms: We were following the convention of previous work (Ladicky et al. CVPR10), who include the parsimony term in the relevance function. However, we will be happy to move it to the diversity section to improve readability. c(l) is assigned the same weight as the diversity term as indicated in L252 (and lambda is learned via cross-val), which guarantees that the positivity condition in L396 holds. We agree that the trade-off between diversity and parsimony can be further explored.
- Hamming Ball Exact Marginal: While we don't have a formal proof, but we believe it's hard to do so. Computing the exact marginal gain is essentially the same problem as computing the size of union of a collection of Hamming balls (centered at the chosen solution in S^(t-1)). This requires a series of terms (of order 1,2..,|S^(t-1)|) as dictated by the inclusion-exclusion principle (http://en.wikipedia.org/wiki/Inclusion%E2%80%93exclusion_principle). The number of terms in this series in exponential in |S^(t-1)|. Our lower-bound terminates the series at the second-order terms.
- MAP results in table 2 are different because of two different approximate MAP solvers: LC/LT use alpha-expansion and HB/DmB use message-passing.
- As described in L315, the accuracy is a recall-based metric (oracle accuracy) to reflect that these solutions will be passed onto some secondary mechanism downstream. A high-recall at small M indicates that this "hypothesis generation" step is working well.
- Supp. Fig 4 bottom: In pascal experiments, for HB, background is treated as just another label. Thus an all black y2 will affect diversity terms (due to Hamming ball centered at all 0). The LC and LT groups are defined only for non-background labels.
- All parameters (HB radius, lambda etc) are chosen via cross-validation.

To summarize, our key contribution is a conceptual connection between submodular maximization on exponentially large item sets and HOPs. While our experiments are only on CV problems, we believe other ML domains such as NLP and speech can also benefit from this connection.